

# Impact of heterozygous hemoglobin E on six commercial methods for hemoglobin A1c measurement

Sharon Yong[1], Hong Liu[1], Cindy Lye Teng Lum[2], Qian Liu[2],
Sin Ye Sim[3], Felicia Fu Mun Chay[3], Wan Ling Cheng[4], Siew Fong Neo[4],
Suru Chew[4], Lizhen Ong[4], Tze Ping Loh[4], Qinde Liu[1], Tang Lin Teo[1]
and Sunil Kumar Sethi[4]

[1] Chemical Metrology Division, Health Sciences Authority, Singapore, Singapore
[2] Department of Pathology, Sengkang General Hospital, Singapore, Singapore
[3] Department of Laboratory Medicine, Alexandra Hospital, Singapore, Singapore
[4] Department of Laboratory Medicine, National University Hospital, Singapore, Singapore

Corresponding author
Tze Ping Loh, tploh@hotmail.com

## ABSTRACT

**Background:** This study examined the impact of heterozygous HbE on HbA1c measurements by six commonly used commercial methods. The results were compared with those from a modified isotope-dilution mass spectrometry (IDMS) reference laboratory method on a liquid chromatograph coupled with tandem mass spectrometer (LC-MS/MS).

**Methods:** Twenty-three leftover samples of patients with heterozygous HbE (HbA1c range: 5.4–11.6%), and nineteen samples with normal hemoglobin (HbA1c range: 5.0–13.7%) were included. The selected commercial methods included the Tina-quant HbA1c Gen. 3 (Roche Diagnostics, Basel, Switzerland), Cobas B 101 (Roche Diagnostics, Basel, Switzerland), D100 (Bio-Rad Laboratories, Hercules, CA, USA), Variant II Turbo HbA1c 2.0 (Bio-Rad Laboratories, Hercules, CA, USA), DCA Vantage (Siemens Healthcare, Erlangen, Germany) and HbA1c Advanced (Beckman Coulter Inc., Brea, CA, USA).

**Results:** With the exception of Cobas B 101 and the Variant II Turbo 2.0, the 95% confidence intervals of the Passing–Bablok regression lines between the results from the six commercial methods and the IDMS method overlapped. The latter suggested no statistically significant difference in results and hence no impact on HbA1c result despite the presence of heterozygous HbE. The method of Cobas B 101 gave positive bias at the range of concentrations examined (5.4–11.6%), while that of Variant II Turbo 2.0 gave positive bias at concentrations up to approximately 9.5%. The finding of significant positive bias in the methods of Cobas B 101 and Variant II Turbo 2.0 agrees with the observations of some previous studies, but is contrary to manufacturer's claim indicating the absence of interference by heterozygous HbE. Our results also clearly showed the impact of heterozygous HbE across a fairly broad measurement range using a laboratory method (the Variant II Turbo 2.0). Laboratory practitioners and clinicians should familiarize themselves with prevailing hemoglobin variants in the population they serve and select the appropriate methods for HbA1c measurement.

## INTRODUCTION

Hemoglobin E (HbE) is a variant hemoglobin caused by a single point mutation, resulting in glutamic acid to lysine substitution at position 26 of the beta chain of the hemoglobin (β 26 Glu→Lys). The amino acid substitution shifts the overall molecular charge more basic, which can be detected by separative methods, such as electrophoresis and liquid chromatography. Globally, HbE is the second most common hemoglobin variant with a prevalence of up to 40% in certain populations in South and South-East Asia (*Bachir & Galacteros, 2004*).

Hemologbin A1c (HbA1c) measurement is recommended for monitoring of long-term glycemic control and treatment titration in patients with diabetes (*American Diabetes Association, 2009*). The measurement of HbA1c can be affected by the presence of hemoglobin variants, leading to spurious measurement that can adversely affect clinical decision making. This study examined the impact of heterozygous HbE on HbA1c measurements using six commonly used commercial methods and a modified isotope-dilution mass spectrometry (IDMS) reference method. The IDMS reference method has demonstrated comparability to the IFCC network (*Liu et al., 2015*).

## MATERIALS AND METHODS

### Study subjects

Twenty-three de-identified leftover whole blood samples from patients who had heterozygous HbE (AE) identified by capillary zone electrophoresis (Capillarys Hemoglobin; Sebia, Cedex, France) and gel electrophoresis (Hydragel Hemoglobin; Sebia, Cedex, France) were included in this study (HbA1c range: 5.4–11.6%, average 7.5%, determined by IDMS). Another twenty de-identified whole blood samples belonging to patients with normal hemoglobin were included as controls (HbA1c range: 5.0–13.7%, average 8.1%, determined by IDMS). The study subjects included patients with diabetes and individuals undergoing wellness screening to allow inclusion of HbA1c samples spanning across the analytical measurement range from both HbE and the normal hemoglobin subjects.

### Ethics declaration

This study was performed as part of the laboratory quality assurance system. The study protocol complies with local regulatory requirements and the Declaration of Helsinki. It has been approved by the institutional ethics review board (National Healthcare Group Domain Specific Review Board, Ref: 2017/00257) with an exemption for written consent for the use of de-identified leftover samples for this study.

### Laboratory analysis

The blood samples were subjected to HbA1c measurement using the Tina-quant HbA1c Gen. 3 (immunoassay adapted on Cobas 501 analyser; Roche Diagnostics, Basel, Switzerland), Cobas B 101 (point-of-care immunoassay, Roche Diagnostics, Basel,

**Table 1 Summary of analytical characteristics of the methods of six commercial methods compared in this study.**

| Name of test | Manufacturer | Test principle | Analytical measurement range in NGSP unit (%) | Published information on interference with HbE |
|---|---|---|---|---|
| Tina-Quant Hemoglobin AIc Gen. 3 | Roche Diagnostics | HbA1c determined by turbidimetric immunoinhibition. Hemoglobin determined by photometric determination after conversion to a colored derivate. | 4.3–18.8 | No interference |
| Cobas B 101 | Roche Diagnostics | HbA1c determined by turbidimetric immunoinhibition. Hemoglobin determined by photometric determination after conversion to a colored derivate. | 4.0–12.0 | No interference |
| D100 | Bio-Rad Laboratories | Cation-exchange high performance liquid chromatography | 3.4–20.6 | No interference |
| Variant II Turbo HbA1c 2.0 | Bio-Rad Laboratories | Cation-exchange high performance liquid chromatography | 3.1–18.5 | No interference |
| DCA Vantage | Siemens Healthcare | HbA1c determined by turbidimetric immunoinhibition. Hemoglobin determined by photometric determination after conversion to a colored derivate. | 4.5–12.0 | No interference |
| HbA1c Advanced | Beckman Coulter Inc | HbA1c determined by turbidimetric immunoinhibition. Hemoglobin determined by photometric determination after conversion to a colored derivate. | 4.0–15.0 | No interference |

**Note:**
These information were obtained from the kit insert of the manufacturer and verified by the laboratory. All HbA1c are reported in National Glycohemoglobin Standardization Program (NGSP) units.

Switzerland), D100 (cation-exchange high performance liquid chromatography (CE-HPLC); Bio-Rad Laboratories, Hercules, CA, USA), Variant II Turbo HbA1c 2.0 (CE-HPLC, Bio-Rad Laboratories, Hercules, CA, USA), DCA Vantage (point-of-care immunoassay, Siemens Healthcare GmbH, Erlangen, Germany) and HbA1c Advanced (immunoassay adapted on DxC 700 AU, Beckman Coulter Inc., Miami, FL, USA). All the commercial methods had been certified by the National Glycohemoglobin Standardization Program and their details are summarised in Table 1. The blood samples were also subjected to a previously described IDMS reference method (Liu et al., 2015) using an Agilent 1290 Infinity liquid chromatograph coupled with AB SCIEX 6500+ tandem mass spectrometer.

It has previously been suggested that the glycation rate of HbE is the same as normal hemoglobin (HbA0) since the modification on HbE is far from the site of glycation (Little et al., 2008). The IDMS HbA1c method involved measuring enzymatically digested N-terminal hexapeptides of the β-chain of the hemoglobin using LC-MS/MS and does not distinguish between HbE and HbA0. Hence, the IDMS method is unaffected by HbE, and the HbA1c results from this method were considered the reference values in our study. The HbA1c were also separately measured on the six commercial methods. The blood

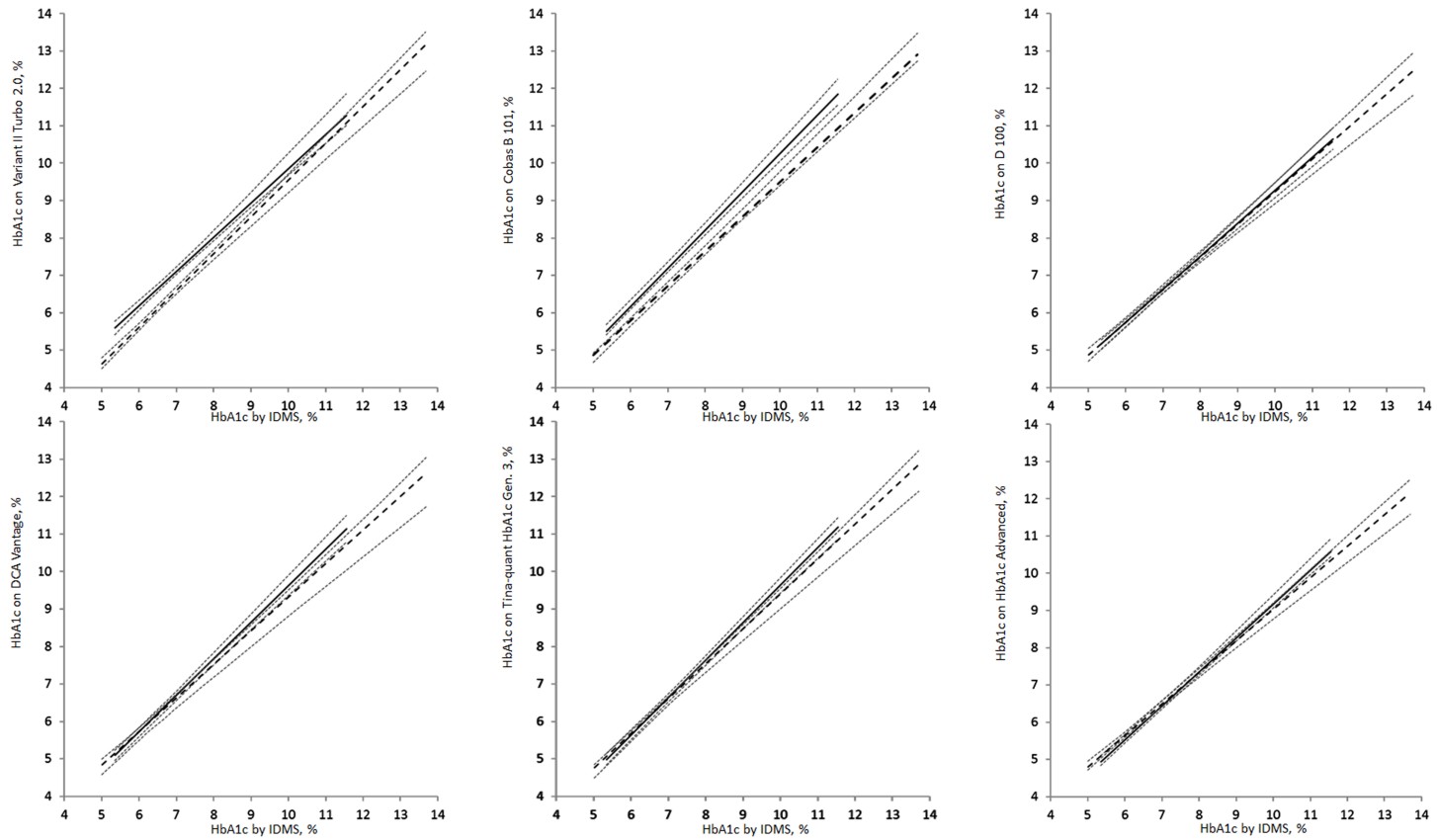

**Figure 1** **Passing–Bablok regression analysis of HbA1c measurements on samples with heterozygous HbE (solid line) and normal hemoglobin (dash lines).** The dotted lines are the 95% confidence intervals of the regression. All HbA1c are reported in National Glycohemoglobin Standardization Program (%) units.

samples were initially analysed by Variant II Turbo 2.0 or Cobas B 101 to ensure that the samples could cover a wide concentration range for this study, then kept at 4 °C for less than 7 days before testing on the other commercial methods. The hemolyzed blood samples were stored at −20 °C for less than 2 months before testing on the LC-MS/MS.

## Statistical analysis

The results for samples with HbE and normal hemoglobin obtained from the methods of the six commercial methods were plotted against those of the IDMS method using Passing–Bablok analysis with 95% confidence intervals obtained by bootstrapping. HbA1c levels of 6% and 9% are clinically important decision values (*Little et al., 2008*). Hence, the relative difference of HbA1c results between samples with HbE and normal hemoglobin were compared. A relative difference of ±6% were considered significant, as recommended by the College of American Pathologists (*College of American Pathologists, 2020*). The statistical analysis was performed using Analyze-It (Microsoft, Redmount, WA, USA).

## RESULTS

The Passing–Bablok regression between the results from the routine methods of six selected commercial methods and IDMS method are shown in Fig. 1. The regression

**Table 2** Regression equations between the commercial methods and the isotope dilution mass spectrometry reference method for samples with heterozygous HbE and normal hemoglobin.

| Routine method | Manufacturer | Slope | Bootstrap 95% CI | Intercept | Bootstrap 95% CI |
|---|---|---|---|---|---|
| **Heterozygous HbE** | | | | | |
| Tina Quant HbA1c Gen 3.0 | Roche Diagnostics | 0.93 | [0.85–1.00] | 0.09 | [−0.54 to 0.56] |
| Cobas B 101 | Roche Diagnostics | 0.93 | [0.90–1.00] | 0.21 | [−0.32 to 0.38] |
| D100 | Bio-Rad Laboratories | 0.87 | [0.78–0.94] | 0.49 | [−0.00 to 1.14] |
| Variant II Turbo HbA1c 2.0 | Bio-Rad Laboratories | 0.98 | [0.88–1.03] | −0.29 | [−0.66 to 0.33] |
| DCA Vantage | Siemens Healthcare | 0.90 | [0.79–0.97] | 0.34 | [−0.14 to 0.96] |
| HbA1c Advance | Beckman Coulter Inc. | 0.85 | [0.77–0.91] | 0.55 | [0.18–1.10] |
| | | | | | |
| **Normal hemoglobin** | | | | | |
| Tina Quant HbA1c Gen 3.0 | Roche Diagnostics | 1.00 | [0.94–1.04] | −0.37 | [−0.71 to 0.07] |
| Cobas B 101 | Roche Diagnostics | 1.02 | [0.97–1.09] | 0.02 | [−0.35 to 0.43] |
| D100 | Bio-Rad Laboratories | 0.88 | [0.83–0.95] | 0.47 | [−0.02 to 0.82] |
| Variant II Turbo HbA1c 2.0 | Bio-Rad Laboratories | 0.92 | [0.85–1.03] | 0.70 | [−0.10 to 1.16] |
| DCA Vantage | Siemens Healthcare | 0.98 | [0.91–1.04] | −0.13 | [−0.56 to 0.32] |
| HbA1c Advance | Beckman Coulter Inc. | 0.91 | [0.87–0.97] | 0.06 | [−0.34 to 0.40] |

**Note:**
The 95% confidence intervals (95% CI) were determined by bootstrapping.

equations are summarized in Table 2. Except for results from Cobas B 101 and the Variant II Turbo 2.0 methods, all the 95% confidence intervals of the regression lines overlapped, suggesting no statistically significant difference in results for the methods of the other four commercial methods. In the presence of heterozygous HbE, the Cobas B 101 method had positive bias at the range of concentrations examined (5.4–11.6%), while the Variant II Turbo 2.0 method had positive bias at concentrations up to approximately 9.5%.

The relative difference between heterozygous HbE and normal hemoglobin at HbA1c of 6% and 9% are summarized in Table 3. Results from the Cobas B 101 method (difference = +6.4% for HbA1c at 6%; difference = +7.4% for HbA1c at 9%) and the Variant II Turbo 2.0 method (difference = +9.9% for HbA1c at 6%) exceeded the a priori criteria for significance. The relative difference at 9% for the Variant II Turbo 2.0 method was +4.4%, which was within the a priori criteria for significance.

## DISCUSSION

The six commercial methods examined in this study included commonly used mainframe analyzer and point-of-care systems. Together, they represented the methods used by 36% of the College of American Pathologists proficiency testing survey participants (*College of American Pathologists, 2020*). This study examined the effect of HbE on HbA1c measurement across the clinically important measurement range by comparing against metrologically traceable reference values determined by a chemical metrology laboratory using IDMS measurements. Using this statistical approach, the effect of HbE on the methods can be examined relative to the normal hemoglobin across a range of concentration, which ameliorates the effects of inter-method calibrator bias.

**Table 3 Difference in HbA1c values.**

| Routine laboratory Method | Manufacturer | At HbA1c = 6% measured by IDMS | | | At HbA1c = 9% measured by IDMS | | |
|---|---|---|---|---|---|---|---|
| | | Normal hemoglobin | Heterozygous HbE | % Difference | Normal hemoglobin | Heterozygous HbE | % Difference |
| Tina Quant HbA1c Gen 3.0 | Roche Diagnostics | 5.6 | 5.7 | −0.9 | 8.6 | 8.5 | 1.7 |
| Cobas B 101 | Roche Diagnostics | 6.2 | 5.8 | **6.4** | 9.2 | 8.6 | **7.4** |
| D100 | Bio-Rad Laboratories | 5.8 | 5.7 | 0.3 | 8.4 | 8.4 | 0.4 |
| Variant II Turbo HbA1c 2.0 | Bio-Rad Laboratories | 6.2 | 5.6 | **9.9** | 8.9 | 8.6 | 4.4 |
| DCA Vantage | Siemens Healthcare | 5.7 | 5.7 | 0.0 | 8.7 | 8.4 | 2.6 |
| HbA1c Advance | Beckman Coulter Inc. | 5.5 | 5.6 | −1.9 | 8.3 | 8.2 | 0.9 |

**Note:**
The values are derived using the regression equations between the commercial methods and the isotope dilution mass spectrometry reference method for samples with heterozygous HbE and normal haemoglobin at 6% and 9% concentrations. All HbA1c are reported in National Glycohemoglobin Standardization Program units. An a priori clinical significance was set at 6% relative difference (values in bold).

The Cobas B 101 method showed significant positive bias across the measurement range investigated in the presence of heterozygous HbE. This finding extended the observation of significant positive bias in the Cobas B 101 method in a previous study that examined only a single patient sample (*Lenters-Westra & Slingerland, 2014*). This finding is contrary to prior published manufacturer's claim, which indicated no significant analytical interference by heterozygous HbE (*Food and Drug Administration, 2020*). The discrepancy may be explained by the use of an acceptability criteria of ±10% relative difference (*Food and Drug Administration, 2020*), which is significantly wider than the clinical standards of ±6% required by the laboratory profession (*College of American Pathologists, 2020*). Laboratory practitioners should carefully communicate the significant positive interference in the Cobas B 101 method due to heterozygous HbE to the clinical user.

The significant positive bias observed in the Cobas B101 method is analytically unexpected. The Cobas B 101 method is a point-of-care assay that applies the principle of turbidimetric immunoinhibition. In general, immunoassays are thought to be more resilient against heterozygous HbE since the amino acid substitution is far from the N-terminus of the β chain where HbA1c glycation and antibody binding occur (*Little et al., 2008*). The cause of this discrepancy remains unclear at present.

On the other hand, the Variant II Turbo 2.0 method showed proportional positive bias in the presence of HbE. This finding is consistent with a previous report by *Sthaneshwar et al. (2013)*. At lower HbA1c concentration, the positive relative bias is large (10%) and reduces at higher HbA1c concentrations. Interestingly, a previous publication reported no significant interference with HbE, despite having 1 out of 11 samples exceeding the acceptability criteria of ±10% relative difference in their internal evaluation (*Food and Drug Administration, 2020*). The positive interference from heterozygous HbE on the Variant II

Turbo 2.0 method may be related to the incomplete cation-exchange chromatography separation. This potentially leads to overlapping HbE and HbA peaks that are sub-optimally resolved by the integration function of the algorithm.

These findings are of clinical concern for two reasons. Firstly, the acceptability criteria used by regulatory body is wider compared to the clinical requirements of routine laboratories. A 10% acceptability criteria implies that a relative difference of such magnitude is clinically acceptable. However, a 10% relative difference in HbA1c value can significantly alter the clinical interpretation of the results of individual patients. At a population level, the mean of HbA1c lies close to the clinical diagnostic threshold of 5.6% for pre-diabetes (*Chai et al., 2017*; *American Diabetes Association, 2009*). The significant bias will lead to over-diagnosis of diabetes, and the consequent unnecessary treatment and erroneous allocation of healthcare resources (*Chai et al., 2017*). As such, regulatory bodies should consider aligning their acceptability criteria with professional bodies to maintain the quality chain from manufacturer to laboratory and clinical end-users.

Secondly, the finding of significant bias meant that individual laboratory must exercise care when selecting a method. Additionally, in regions where a particular hemoglobin variant is prevalent, it may be prudent for the laboratory to verify the prior data, especially when its clinical requirements differ significantly from the manufacturer's acceptability criteria. However, such verification exercise is often beyond the means of individual routine laboratories. This can be overcome by forming a collaborative network of laboratories to jointly evaluate the methods. When such exercise is performed, it is desirable to employ a reference laboratory method that is not influenced by the presence of hemoglobin variant in question.

Nevertheless, access to such reference laboratory method is limited. Increased reporting of hemoglobin variant interference in the literature will help laboratory practitioners make more informed decisions and assist manufacturers in improving their analytical methods. It may be necessary to periodically re-evaluate hemoglobin variant interference as a manufacturer may reformulate their reagents or modify their analytical methods, particularly for the CE-HPLC (*Little & Rohlfing, 2017*).

There is currently no consensus on how laboratory should manage hemoglobin variants that are incidentally detected during HbA1c measurements (*Lewis et al., 2017*). Some hemoglobin variants can interfere with HbA1c measurements leading to spurious results. Others may alter the lifespan of the red blood cells, leading to an altered glycation rate of the hemoglobin. An example of this is homozygous hemoglobin S that causes sickle cell disease. It is associated with shortened red blood cell lifespan, which causes lower HbA1c values relative to the ambience glucose. As such, it is advantageous to communicate the presence of hemoglobin variants to the clinicians to help them optimally interpret the HbA1c values. Additionally, such information also provides an opportunity for work up of significant hemoglobinopathy.

In general, it is desirable to avoid the use of methods that may be significantly interfered by the hemoglobin variant that is prevalent in the population served. This is particularly true for laboratory methods such as CE-HPLC, where the presence of hemoglobin variants may be inferred only by careful manual reading of the chromatograms by skilled

technologists. It is desirable to avoid reporting an HbA1c result with known hemoglobin variant interference. When detected, alternative method not affected by the hemoglobin variant should be used to report the HbA1c instead. Alternate biomarker such as fructosamine may also be considered. However, such workflows require access to alternate instruments or esoteric biomarkers, which may not even be readily available to most laboratories.

The use of non-separative laboratory method such as the immunoassays for HbA1c measurement may mask an undetected interfering hemoglobin variant. At the same time, immunoassays are also liable to interfering antibodies (*Rhea & Molinaro, 2014*). Both of these may lead to erroneous reporting that can evade laboratory detection. The use of devices with such test principles in the point of care setting is considered an increased risk as laboratory supervision may be limited. As such clinicians should remain highly vigilant against clinically discordant HbA1c.

## CONCLUSIONS

Four out of six commercial HbA1c laboratory methods examined in this study showed significant bias in measurement in the presence of heterozygous HbE when compared to an IDMS method. The impact of heterozygous HbE on HbA1c measurement may not be predicted from the assay principle, nor be adequately disclosed by the manufacturer. It is important for clinical laboratory to understand the impact of prevailing hemoglobin variant in their population during method selection and evaluation.

## ACKNOWLEDGEMENTS

Lingkai Wong conducted part of IDMS measurements when he was a staff at Health Sciences Authority.

### Funding
The authors received no funding for this work.

### Competing Interests
The authors declare that they have no competing interests.

### Author Contributions
- Sharon Yong conceived and designed the experiments, performed the experiments, analyzed the data, prepared figures and/or tables, authored or reviewed drafts of the paper, and approved the final draft.
- Hong Liu conceived and designed the experiments, performed the experiments, analyzed the data, prepared figures and/or tables, authored or reviewed drafts of the paper, and approved the final draft.
- Cindy Lye Teng Lum conceived and designed the experiments, performed the experiments, prepared figures and/or tables, and approved the final draft.

- Qian Liu conceived and designed the experiments, performed the experiments, prepared figures and/or tables, and approved the final draft.
- Sin Ye Sim conceived and designed the experiments, performed the experiments, prepared figures and/or tables, and approved the final draft.
- Felicia Fu Mun Chay conceived and designed the experiments, performed the experiments, prepared figures and/or tables, and approved the final draft.
- Wan Ling Cheng conceived and designed the experiments, performed the experiments, prepared figures and/or tables, and approved the final draft.
- Siew Fong Neo conceived and designed the experiments, performed the experiments, prepared figures and/or tables, and approved the final draft.
- Suru Chew conceived and designed the experiments, performed the experiments, prepared figures and/or tables, and approved the final draft.
- Lizhen Ong conceived and designed the experiments, performed the experiments, prepared figures and/or tables, and approved the final draft.
- Tze Ping Loh conceived and designed the experiments, analyzed the data, prepared figures and/or tables, authored or reviewed drafts of the paper, and approved the final draft.
- Qinde Liu conceived and designed the experiments, analyzed the data, prepared figures and/or tables, authored or reviewed drafts of the paper, and approved the final draft.
- Tang Lin Teo conceived and designed the experiments, analyzed the data, prepared figures and/or tables, authored or reviewed drafts of the paper, and approved the final draft.
- Sunil Kumar Sethi conceived and designed the experiments, analyzed the data, prepared figures and/or tables, and approved the final draft.

### Ethics

The following information was supplied relating to ethical approvals (i.e., approving body and any reference numbers):

The NHG Domain Specific Review Board (DSRB) approved this research (NHG DSRB Ref: 2017/00257).

### Data Availability

Raw data are available as a Supplemental File.

### Supplemental Information

Supplemental information for this article can be found online at http://dx.doi.org/10.7717/peerj-achem.9#supplemental-information.

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
