# Peer review of "Impact of heterozygous hemoglobin E on six commercial methods for hemoglobin A1c measurement"

_PeerJ Analytical Chemistry, doi:10.7717/peerj-achem.9_

## Round 0.1 · original submission · Minor Revisions

Please address all the comments provided by the reviewers.

·

Basic reporting

no comment

Experimental design

no comment

Validity of the findings

no comment

Additional comments

General comments: The authors examine interference from HbAE using six commercial HbA1c methods. They compared results with an LC-MS/MS. There were no statistically difference results for HbAE for 4 of these methods. However, for two methods (VII Turbo 2.0 and Cobas b101) there were statistically significant differences despite manufacturer’s claim s to the contrary. This is an important paper that adds to the information we have on variant interference with HbA1c methods. This is especially important since there are discordant data in the literature for the VII T2.0 on this interference.

Specific comments:

1. The authors should clarify up front whether or not the LC-MS/MS method they use as a reference method is part of the IFCC network. If not, they need to show some data to indicate comparability with the IFCC network, which is monitored on a regular basis.
2. Line 126-127: 36% seems a bit high. I calculate ~31% of participants.
3. Table3: it would be helpful to highlight those over 6%.
4. Figure 1: label x-axes
5. Figure 1: would it be possible to include the data points so that the distribution of points can be seen.

Reviewer 2 ·

Basic reporting

The authors wrote a clear manuscript with acceptable English and well-organized structure. However, the title of x-axis is required in fugure 1.

Experimental design

If possible, the average of blood sugar levels from both EA and A2A groups should provide in the manuscript. The authors should specify the type of patients such as diabetic patients for replacing patients in the study subjects.

Validity of the findings

No comment

Additional comments

The authors of the manuscript report the impact of Hb E heterozygote on 6 commercial methods for Hb A1c measurement and IDMS was used as reference laboratory method. EA and A2A Hb types with different levels of Hb A1c were used to determined Hb A1c levels by 6 selected commercial methods. The manuscript is well written. However, there are some points shoud be addressed before publication.

---

## Round 0.2 · Minor Revisions

Please consider the most recent comments provided by the reviewers.

·

Basic reporting

acceptable

Experimental design

acceptable

Validity of the findings

acceptable

Additional comments

the authors have addressed all of my concerns except that I still think that the X-axes in figure 1 should be labeled within the figure itself. I will leave this up to the editors.

Reviewer 2 ·

Basic reporting

No comment

Experimental design

No comment

Validity of the findings

According to the objective of this study, this study examined the impact of heterozygous HbE on HbA1c measurements using six commonly used commercial methods and a modified isotope dilution mass spectrometry (IDMS) reference method, the authors need to revise the conclusions for linking to original research question.

Additional comments

There is minor revision of the manuscript to be considered before further publication.

---

## Round 0.3 · accepted · Accept

The authors have addressed the reviewer's comments.